# Prediction of femoral osteoporosis using machine-learning analysis with radiomics features and abdomen-pelvic CT: A retrospective single center preliminary study

Hyun Kyung Lim[1], Hong Il Ha[2]*, Sun-Young Park[2], Junhee Han[3]

**1** Department of Radiology, Soonchunhyang University Seoul Hospital, Seoul, Republic of Korea,
**2** Department of Radiology, Hallym University Sacred Heart Hospital, Anyang-si, Gyeonggi-do, Republic of Korea, **3** Department of Statistics and Data Science Convergence Research Center, Hallym University, Chuncheon-si, Gangwon-do, Republic of Korea

* ha.hongil@gmail.com

## Abstract

### Background

Osteoporosis has increased and developed into a serious public health concern worldwide. Despite the high prevalence, osteoporosis is silent before major fragility fracture and the osteoporosis screening rate is low. Abdomen-pelvic CT (APCT) is one of the most widely conducted medical tests. Artificial intelligence and radiomics analysis have recently been spotlighted. This is the first study to evaluate the prediction performance of femoral osteoporosis using machine-learning analysis with radiomics features and APCT.

### Materials and methods

500 patients (M: F = 70:430; mean age, 66.5 ± 11.8yrs; range, 50–96 years) underwent both dual-energy X-ray absorptiometry and APCT within 1 month. The volume of interest of the left proximal femur was extracted and 41 radiomics features were calculated using 3D volume of interest analysis. Top 10 importance radiomic features were selected by the intraclass correlation coefficient and random forest feature selection. Study cohort was randomly divided into 70% of the samples as the training cohort and the remaining 30% of the sample as the validation cohort. Prediction performance of machine-learning analysis was calculated using diagnostic test and comparison of area under the curve (AUC) of receiver operating characteristic curve analysis was performed between training and validation cohorts.

### Results

The osteoporosis prevalence of this study cohort was 20.8%. The prediction performance of the machine-learning analysis to diagnose osteoporosis in the training and validation cohorts were as follows; accuracy, 92.9% vs. 92.7%; sensitivity, 86.6% vs. 80.0%; specificity, 94.5% vs. 95.8%; positive predictive value, 78.4% vs. 82.8%; and negative predictive value, 96.7% vs. 95.0%. The AUC to predict osteoporosis in the training and validation

**Data Availability Statement:** The datasets in this paper can be fully accessed through Zenodo: https://doi.org/10.5281/zenodo.4460972.

**Funding:** This study was supported by the Soonchunhyang University Research fund and the DongKook Life Science. Co., Ltd., Republic of Korea (DK-IIT2019-03). The DongKook Life Science. Co., Ltd. has no competing interests relating to employment, consultancy, patents, products in development, marketed products, etc. These funders had no role in study design, data collection and analysis, decision to publish, or preparation of the manuscript.

**Competing interests:** The authors have no competing interests to declare related to the DongKook Life Science. Co., Ltd. This does not alter our adherence to PLOS ONE policies on sharing data and materials.

**Abbreviations:** APCT, abdomen-pelvic CT; AUC, area under the curve; BMD, bone mineral density; CTHU, CT Hounsfield unit; DXA, dual-energy X-ray absorptiometry; HU, Hounsfield unit; HUHA, Hounsfield unit histogram analysis; ICC, intraclass correlation coefficient; RF, random Forest; VOI, volume of interest.

cohorts were 95.9% [95% confidence interval (CI), 93.7%-98.1%] and 96.0% [95% CI, 93.2%-98.8%], respectively, without significant differences (P = 0.962).

## Conclusion

Prediction performance of femoral osteoporosis using machine-learning analysis with radiomics features and APCT showed high validity with more than 93% accuracy, specificity, and negative predictive value.

## Introduction

As the elderly population has rapidly grown, osteoporosis has increased and developed into a serious public health concern [1]. Approximately 30% of all postmenopausal women have osteoporosis in the developed countries, and up to 50% of these patients will sustain one or more osteoporotic fracture in their life time [2]. Although the prevalence of osteoporosis is very high, it has specific diagnosis tool such as dual-energy X-ray absorptiometry (DXA), effective treatment options, and preventive methods [1]. Therefore, osteoporosis is a disease in which screening can have a great effect on patient outcomes [3]. However, screening for osteoporosis using DXA has been underperformed because osteoporosis is asymptomatic until major incidental fragile fractures occur, such as vertebral body or hip fractures [2]. Patients often do not recognize the seriousness of this disease and, therefore, do not participate in the screening program voluntarily [4]. There is a growing consensus regarding the need for alternative screening methods to overcome the limitations and underuse of DXA as a screening method for osteoporosis. Abdomen-pelvic computed tomography (APCT) is commonly performed on adults to evaluate various diseases, during routine health check-ups or follow-up diagnosed diseases. Even if a small number of these scans were used to opportunistically screen for osteoporosis, the impact could be substantial. Several studies have shown optimistic results using APCT for opportunistic screening for osteoporosis [5–7].

Radiomics is the most advanced application within the radiology research field. It extracts various features from medical images and has the potential to find disease characteristics that fail to be appreciated by the naked eye using specially designed data-characterization algorithms for image analysis [8]. These radiomics features are the distinctive imaging features between disease forms might be useful for predicting prognosis and therapeutic response for various conditions, thus providing valuable information for personalized therapy [9, 10]. As osteoporosis progresses, bone mineral density (BMD) is decreased and bony microstructure change occur, simultaneously [11]. BMD-decrease is highly correlated with mean computed tomography Hounsfield unit (CTHU) change, as well as HU histogram analysis (HUHA) of the proximal femur volume representing fatty marrow content ($HUHA_{Fat}$, percentage ratio of HU range $\leq$ 0HU) or thick cortical bone content ($HUHA_{Bone}$, percentage ratio of HU range $\geq$126HU) [5, 12]. This radiomics analysis may be useful for the evaluation of microstructure changes of trabecular bone [13–15]. Additionally, machine-learning analysis is changing the paradigm of medical practice, which is best suited for mass screening [16]. However, to the best of our knowledge, there has been no research evaluating femoral osteoporosis using radiomics features and machine learning analysis. Therefore, the purpose of this study was to evaluate the predicting performance of machine learning analysis for diagnosing femoral osteoporosis using radiomics features and APCT.

## Materials and methods

This retrospective study was approved by institutional review board and ethics committee at Hallym University Sacred Heart hospital and the need for informed consent was waived.

### Patients

Between July 2018 and June 2019, 569 patients aged 50 years or older who had undergone APCT and DXA within an interval of a 1-month period (mean, 3.8 ± 6.1 days; range, 0–30 days) were retrospectively included. Among these patients, 69 were excluded due to bone metastases (n = 8), metastasis other than bone (n = 10), history of receiving chemotherapy within the last 3 months (n = 26), primary bone disease (e.g., fibrous dysplasia; n = 4), developmental or traumatic deformation of the femur (n = 6), or any total hip arthroplasty or internal nailing (n = 15). Finally, 500 patients (mean age, 66.5 ± 11.8 yrs; range, 50–96 years) were included. This cohort consisted of 70 men (mean age, 72.6 ± 8.0 yrs; range, 54–89 years) and 430 women (mean age, 65.4 ± 12.1 yrs; range, 50–96 years). There were no duplicate patients enrolled. The reasons for CT imaging were as follows: cancer metastasis surveillance (n = 327), minor trauma (e.g., slip-down injury or simple fall-down injury; n = 37), or routine health check-up or medical inspection (n = 136). Among the included patients, 70% were randomly selected to be the training cohort (M:F = 48:302; age, 66.8±12.2 yrs), and the remaining 30% were selected to be the validation cohort (M:F = 22:128; age, 65.7±11.1 yrs) (Fig 1).

### Dual-energy X-ray absorptiometry

DXA of the proximal femur for BMD assessment was performed using a single BMD scanner (GE Healthcare Lunar Prodigy Densitometers, Madison, WI, USA). The lowest $T$-score of the femoral neck was used as the reference standard. Osteoporosis of the femur was defined as a $T$-score $\leq -2.5$, and non-osteoporosis was defined as a $T$-score $> -2.5$ (1).

### Computed tomography imaging

All CT examinations were performed using two multidetector-row CT scanners (SOMATOM Definition Edge, SOMATOM Definition Flash; Siemens Healthcare, Forchheim, Germany) in the standard single-energy CT mode. Automatic tube voltage selection (Care kVp) and automatic tube current modulation (CARE Dose 4D) protocols were applied. To exclude the effect of the contrast agent on the CTHU measurement, all measurements were performed on only pre-contrast CT scans [17]. The scanning parameters were as follows: detector collimations, $128 \times 0.6$ mm; pitch, 0.6; gantry rotation time, 0.5 s; tube current, 200 or 289 mAs; tube voltage, 100 or 120 kVp; and iterative reconstruction (sinogram-affirmed iterative reconstruction, S1, I40f). The voxel size of all raw data was 0.67mm x 0.67mm x 1mm.

### Radiomics analysis

The radiomics analysis was performed using two commercial software programs (Aquarius iNtuition v4.4.12®, TeraRecon, Foster City, CA, USA; Medip®, Medical Imaging Solution for Segmentation and Texture Analysis, Korea). Each target volume of interest (VOI) of the left femur was extracted using the region growing editing tool. The area below the lesser trochanter of the femur was excluded to maintain a constant VOI across all measurement (Fig 2). For each VOI, 41 radiomic features were extracted and divided into four groups: (1) first-order grey-level histogram features to describe the distribution of grey-values within the volume; (2) geometric features to describe the shape and size of the volume of interest; (3) grey-level co-occurrence level matrices are statistical features used to explore the spatial relationship

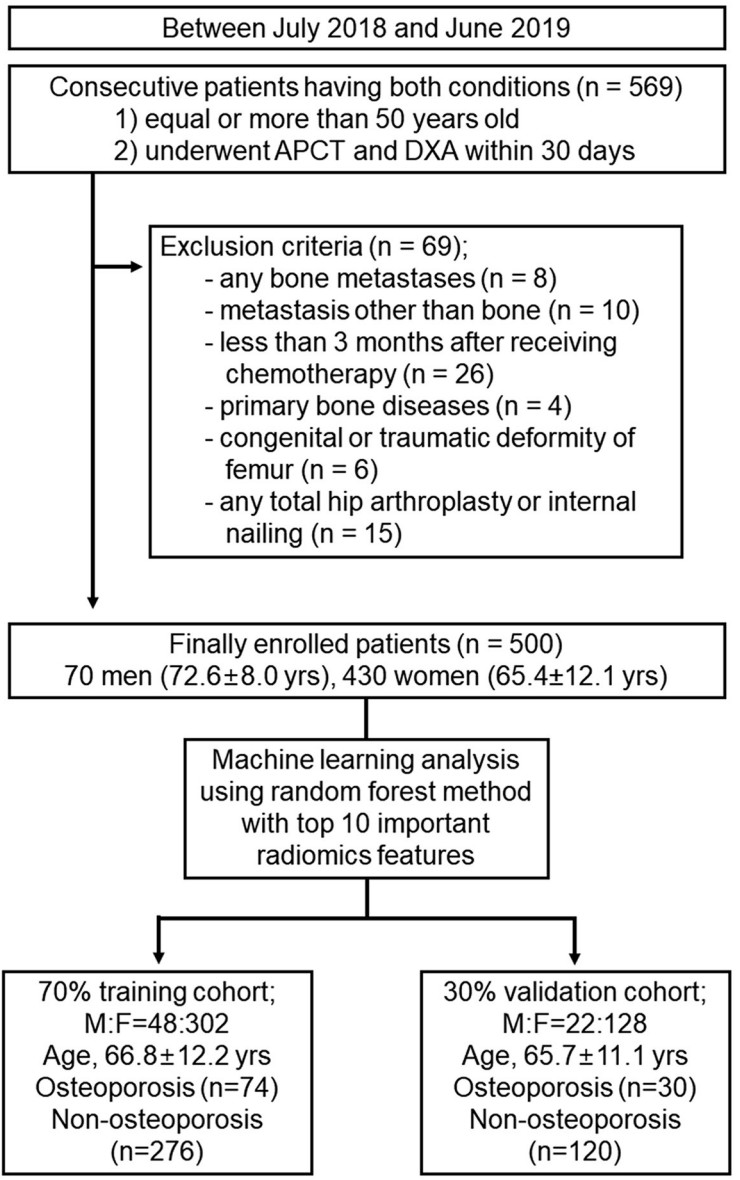

**Fig 1. Flowchart of patient enrollment and random selection for machine-learning analysis.**

between two pixels with certain distance and direction, including contrast relation, entropy, angular second moment, etc.; and (4) wavelet transformation is a transformation that separates data into different frequency components, and then examines each component with resolution matched to its scale. A detailed description of these radiomics features are independent of imaging modality and can be found in the literature [18–20].

## Feature extraction and random forest model

Our study design followed the Transparent Reporting of a multivariable prediction model for Individual Prognosis or Diagnosis (TRIPOD) guidelines [21]. To overcome the repeatability weakness of radiomics, only those with an intraclass correlation coefficient (ICC) value higher than 0.9 were considered stable and selected for subsequent analysis. We used the random

(a)

(b)

**Fig 2. Measurement of the 3D volume of proximal femur.** (A) Texture analysis. Forty-one features are extracted from the volume of interest (red). (B) Mean CTHU and HU histogram analysis (HUHA) are simultaneously calculated from the other commercial 3D image processing software. Negative HU range (yellow box, $HUHA_{Fat}$, $\leq 0HU$) is considered as fatty marrow content and equal or more than 126 HU range (red box, $HUHA_{Bone}$) is considered as dense bone content, respectively. (A) Texture analysis. (B) Mean-CTHU and HU histogram analysis (HUHA).

forest (RF) algorithm of various machine-learning analysis methods. Although the RF algorithm itself enables the efficient selection of the highly correlated variables and reduces the number of variables, further feature selection was performed by the Mean Decrease in Gini index, and the top 10 important features were selected (Fig 3) [22]. The RF algorithm is an ensemble of unpruned classification or regression trees created by using bootstrap samples of the training data and random feature selection in tree induction. Predictions are made by aggregating the predictions of the ensemble. We randomly selected 70% (n = 350) of the samples as the training cohort and the remaining 30% (n = 150) as the validation cohort using a 'caret' R package [23]. An RF is a meta-estimator that fits a number of decision-tree classifiers on various sub-samples of the dataset. After each tree, the decision for classification result was determined, and the result was averaged to improve the predictive accuracy and control overfitting. In the RF algorithm, each tree in the ensemble is built from a sample drawn with a replacement from the training cohort. In addition, when splitting a node during the construction of the tree, the split that is picked is the best split among a random subset of the features. For the best hyperparameter tuning, 5-fold cross validation using random search was performed and the result was summarized in S1 Table. The hyperparameter of our RF algorithm were as follows: mtry = 3, minimum nodal size = 11, and splitrule = extratrees. The best model was selected and validated in the test cohort. Further explanation of RF model and features was summarized in S1 Appendix.

## Statistical analysis

All statistical analyses were performed with the open-source statistical computing environment R (version 3.6.1; R Foundation for Statistical Computing) and the Medcalc Statistical Software version 19.1.3 (MedCalc Software bv, Ostend, Belgium). The reproducibility of the radiomics features was evaluated by ICC using a two-way random model with absolute measurements. To assess ICC, two radiologists (one with 12 years of experience interpreting body images and the other with 6 years of experience interpreting musculoskeletal images) measured the radiomics features in images from 40 randomly selected cases. Using the diagnostic test confusion matrix, the prediction accuracy of the training and validation cohorts were calculated. The area under curve (AUC) of the receiver operating characteristic curve and 95% confidence interval (CI) of the training and validation cohorts were calculated. The AUCs of training and validation models were compared using the method developed by DeLong et al. A $P$-value $< 0.05$ was considered a significant difference.

## Results

The demographic information of the study cohort is summarized in Table 1. Overall, 104 patients were diagnosed with osteoporosis, and the osteoporosis prevalence of this cohort was 20.8%.

The AUC and correlation coefficient to predict femoral osteoporosis and the ICC of all radiomics features are summarized in Table 2. According to the RF feature selection algorithm made by the mean decrease in Gini index and distribution of minimal depth, wavelet$_{LLL}$, HUHA$_{Bone}$, mean-CTHU, HUHA$_{Fat}$, wavelet$_{HLL}$, kurtosis, wavelet$_{LLH}$, wavelet$_{LHL}$, texture energy, moment, and skewness were selected as the top important radiomics features (Fig 3).

The prediction accuracy of the machine-learning analysis using the RF model to diagnose osteoporosis in training and validation cohorts are summarized in Table 3. Both cohorts showed more than 80% of sensitivity, 94% of specificity, 94% of negative predictive value (NPV), and 93% of accuracy. Training model showed 95.9% of AUC (95% CI, 93.7%-98.1%) and validation model showed 96.0% of AUC (95% CI, 93.2%-98.8%). There was no significant

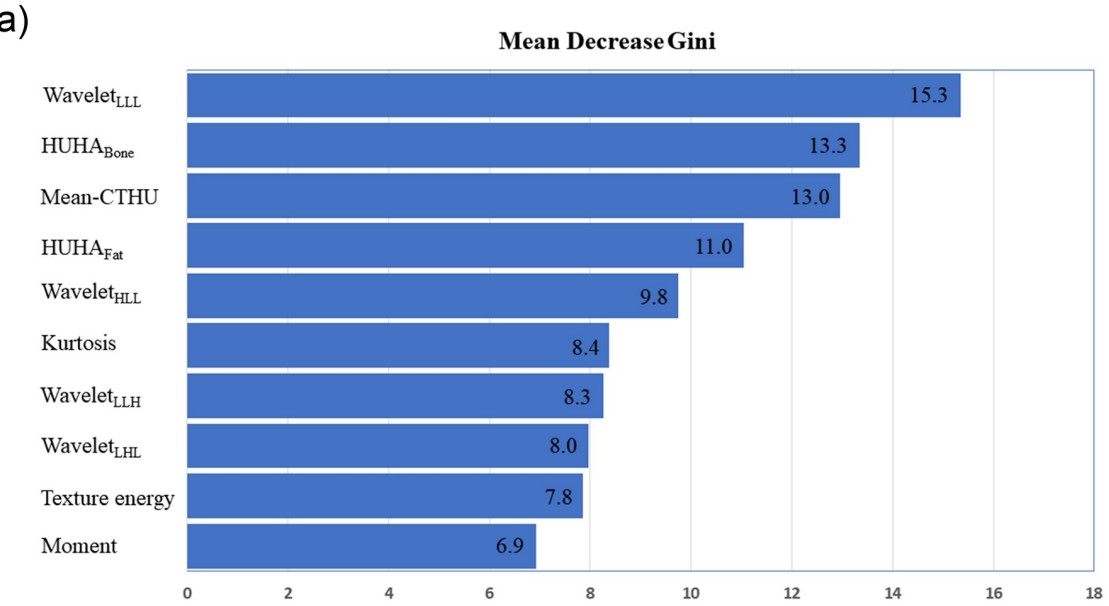

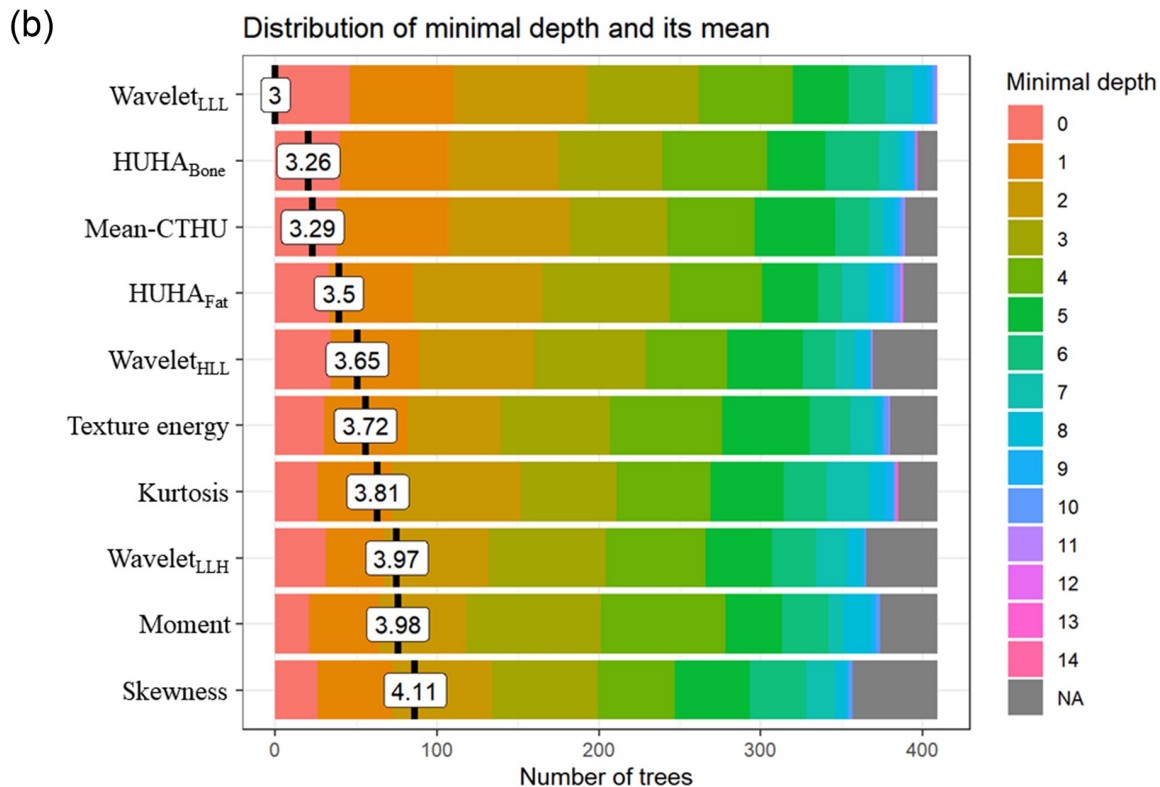

**Fig 3. Top 10 important features with mean decrease in Gini index (A) and distribution of minimal depth (B).** A higher mean decrease in Gini index indicates higher features importance. The distribution of minimal depth is marked by a vertical bar with a value label on the trees of random forest. (A) Mean decrease in Gini index. (B) Distribution of minimal depth.

**Table 1. Comparison of demographics between the osteoporosis and non-osteoporosis groups.**

| | Osteoporosis (n = 104) | Non-osteoporosis (n = 396) | *P*-value |
|---|---|---|---|
| **Sex (M:F)** | 8:96 | 62:334 | < 0.001 |
| **Age (years, mean ± SD)** | 78.6 ± 8.6 | 63.3 ± 10.5 | < 0.000 |
| **T-score** | -3.11 ± 0.53 | -0.80 ± 1.01 | < 0.000 |
| **BMD (g/cm$^2$)** | 0.57 ± 0.06 | 0.84 ± 0.12 | < 0.000 |
| **BMI (kg/m$^2$)** | 22.1 ± 3.7 | 24.4 ± 4.1 | < 0.000 |
| **Interval between DXA to APCT (day)** | 6.5 ± 7.2 | 3.1 ± 5.5 | < 0.000 |

APCT = abdominal-pelvic CT; BMD = bone material density; BMI, body mass index; DXA = dual-energy X-ray absorptiometry.

difference of AUCs predicting femoral osteoporosis between training and validation models (*P* = 0.962) (Fig 4).

## Discussion

The primary goal of this study was to evaluate the prediction accuracy of osteoporosis using machine-learning analysis with radiomics features and APCT. In this study, the prediction accuracy of osteoporosis was 95.9% and 96.0% in the training and validation cohorts, respectively. The predicting performance for diagnosis of osteoporosis was 95.8% in specificity, 95% in NPV, 80% in sensitivity, and 92.7% in diagnostic accuracy at validation cohort. In particular, our results showed high specificity and NPV more than 95%, which is considered as meaningful results to select healthy peoples. Therefore, screening using this method may contribute to reduce or prevent the unnecessary duplication check and cost of DXA.

In order to increase the osteoporosis screening rate, several policies have been tried such as patient selection using questionnaires, education of primary clinic physicians, coverage by medical insurance, and so on [24–29]. However, more than 25 million abdominal-pelvic CT scans have been performed on adults each year in the United States. If osteoporosis screening with APCT is possible, the effect would be enormous. Based on this concept, several studies have shown optimistic results. Most studies have reported the usefulness of osteoporosis diagnosis by mean CTHU measurement [5–7, 30]. In some studies, images were analyzed using HU histogram analysis and texture analysis [5, 31, 32]. Recently, a few studies on the usefulness of osteoporosis screening using artificial intelligence have been published [6, 33] [new ref DII]. The advantage of precision medicine using artificial intelligence is that auto-segmentation and mass-screening using big data is possible (10, 11). In this study, femur segmentation was performed by researchers using a semi-automatic extraction using 3D image analysis software. However, the auto-segmentation algorithm of the femur has been recently developed and applied to image analysis for research purpose. If actual clinical application is made, osteoporosis screening can be easier and more effective and may improve patient convenience. Although there may be concerns about radiation exposure, the concept of opportunistic screening using APCT is to obtain additional information related to osteoporosis from already-performed CT scans for other medical reasons. As a result, patients can simultaneously obtain secondary bone health information without additional radiation exposure, thereby gaining additional benefits in terms of cost, time, and convenience.

We used top 10 radiomics features based on the RF feature selection. Among them, mean-CTHU is the typical feature in conventional CT image analysis. This conventional feature was selected as a meaningful feature with high feature importance scores. HUHA$_{Fat}$ and HUHA$_{Bone}$ features are based on the HU number distribution and represented specific tissue contents such as fatty marrow and dense bone content, respectively. Kurtosis and skewness measure the

**Table 2. Summary of AUC and correlation coefficient to predict femoral osteoporosis, and intraclass correlation coefficient of all radiomics features.**

| Radiomic Feature | AUC (95% CI) | Correlation coefficient | ICC (95% CI) |
|---|---|---|---|
| **First-order gray-level histogram (n = 9)** | | | |
| Entropy | 0.884 (0.853–0.911) | -0.573 | 0.993 (0.987–0.996) |
| HUHA_bone | 0.950 (0.927–0.967) | -0.327 | 0.999 (0.999–0.999) |
| HUHA_fat | 0.938 (0.908–0.953) | 0.699 | 0.997 (0.995–0.998) |
| Kurtosis | 0.903 (0.873–0.927) | 0.621 | 0.993 (0.988–0.996) |
| Mean_CTHU | 0.951 (0.928–0.968) | -0.628 | 0.989 (0.903–0.996) |
| Skewness | 0.876 (0.844–0.903) | 0.544 | 0.991 (0.984–0.994) |
| Texture_energy | 0.877 (0.845–0.904) | -0.470 | 0.987 (0.973–0.992) |
| Uniformity | 0.866 (0.833–0.895) | 0.549 | 0.994 (0.990–0.996) |
| Variance | 0.852 (0.817–0.882) | -0.505 | 0.981 (0.966–0.989) |
| **Geomatric features (n = 6)** | | | |
| Discrete Compactness | 0.562 (0.517–0.606) | 0.075 | 0.979 (0.962–0.987) |
| Effective diameter | 0.501 (0.456 to 0.545) | -0.010 | 0.934 (0.882–0.960) |
| Roundness | 0.723 (0.682–0.762) | -0.325 | 0.953 (0.917–0.973) |
| Sphericity | 0.570 (0.523–0.612) | -0.140 | 0.977 (0.958–0.986) |
| Texture_compactness1 | 0.582 (0.538–0.626) | -0.113 | 0.881 (0.795–0.931)* |
| Texture_compactness2 | 0.701 (0.659–0.741) | -0.292 | 0.843 (0.738–0.908)* |
| **Co-occurrence matrix (n = 18)** | | | |
| CROSS_GLCMASM | 0.776 (0.737–0.812) | 0.466 | 0.993 (0.986–0.995) |
| CROSS_GLCMcontrast | 0.683 (0.661–0.743) | -0.1226 | 0.969 (0.942–0.983) |
| CROSS_GLCMentropy | 0.821 (0.766–0.837) | -0.452 | 0.987 (0.974–0.993) |
| CROSS_GLCMIDM | 0.662 (0.619–0.703) | 0.229 | 0.998 (0.995–0.998) |
| EW_GLCMASM | 0.782 (0.744–0.818) | 0.467 | 0.992 (0.985–0.995) |
| EW_GLCMcontrast | 0.667 (0.645–0.728) | -0.119 | 0.960 (0.926–0.977) |
| EW_GLCMentropy | 0.775 (0.754–0.827) | -0.438 | 0.987 (0.974–0.993) |
| EW_GLCMIDM | 0.682 (0.639–0.723) | 0.270 | 0.998 (0.995–0.998) |
| Homogeneity | 0.787 (0.748 to 0.822) | 0.409 | 0.996 (0.993–0.998) |
| Moment | 0.757 (0.716–0.794) | 0.446 | 0.980 (0.965–0.988) |
| NS_GLCMASM | 0.760 (0.721–0.797) | 0.446 | 0.992 (0.985–0.995) |
| NS_GLCMcontrast | 0.692 (0.668–0.749) | -0.122 | 0.978 (0.956–0.988) |
| NS_GLCMentropy | 0.757 (0.734–0.809) | -0.408 | 0.983 (0.966–0.991) |
| NS_GLCMIDM | 0.640 (0.596–0.682) | 0.182 | 0.998 (0.995–0.998) |
| SIX_GLCMASM | 0.803 (0.765–0.837) | 0.495 | 0.993 (0.987–0.996) |
| SIX_GLCMcontrast | 0.769 (0.729–0.805) | -0.244 | 0.966 (0.935–0.981) |
| SIX_GLCMentropy | 0.835 (0.800–0.867) | -0.500 | 0.988 (0.974–0.993) |
| SIX_GLCMIDM | 0.686 (0.643–0.726) | 0.294 | 0.998 (0.995–0.998) |
| **Wavelet transformation (n = 8)** | | | |
| Wavelet_HHH | 0.626 (0.582–0.668) | -0.175 | 0.379 (0.109–0.596)* |
| Wavelet_HHL | 0.795 (0.757–0.830) | -0.397 | 0.642 (0.444–0.780)* |
| Wavelet_HLH | 0.767 (0.728–0.803) | -0.366 | 0.618 (0.410–0.765)* |
| Wavelet_HLL | 0.927 (0.900–0.948) | -0.583 | 0.966 (0.940–0.980) |
| Wavelet_LHH | 0.731 (0.690–0.769) | -0.334 | 0.607 (0.395–0.757)* |
| Wavelet_LHL | 0.915 (0.887–0.938) | -0.565 | 0.956 (0.923–0.975) |
| Wavelet_LLH | 0.912 (0.884–0.935) | -0.550 | 0.949 (0.910–0.970) |
| Wavelet_LLL | 0.950 (0.928–0.968) | -0.640 | 0.999 (0.998–0.999) |

Values in parentheses mean 95% confidential interval.

* Values are excluded in random forest analysis.

**Table 3. Diagnostic performance of the Machine-learning analysis.**

| | Training cohort (n = 350) | Validation cohort (n = 150) |
|---|---|---|
| Sensitivity (%) | 86.6 (76.1–93.7) | 80.0 (61.4–92.3) |
| Specificity (%) | 94.5 (91.0–96.7) | 95.8 (90.5–98.6) |
| Positive likelihood ratio | 15.3 (9.4–24.9) | 19.2 (8.0–46.1) |
| Negative likelihood ratio | 0.1 (0.1–0.3) | 0.2 (0.1–0.43) |
| Disease prevalence (%) | 19.1 | 20.0 |
| Positive predictive value (%) | 78.4 (69.1–85.5) | 82.8 (66.6–92.0) |
| Negative predictive value (%) | 96.7 (94.2–98.1) | 95 (90.4–97.5) |
| Accuracy (%) | 92.9 (89.6–95.3) | 92.7 (87.3–96.3) |

‡Values in parentheses mean 95% confidential interval.

peakedness and symmetry of the HU histogram. The effective diameter was defined as the diameter of a sphere whose volume is equal to the segmented volume. The wavelet features are image transform technique based on the space–frequency decomposition with low computational complexity [18, 19]. In addition to the mean-CTHU as a conventional feature, these radiomic features were selected as important features evaluating osteoporosis and thought to reflect the pathophysiology of osteoporosis not detectable by naked human eyes.

We analyzed the left femur from the head to lesser trochanter as a VOI because this area is consistent with the DXA target range. As the proximal femur has a 3D complex structure, 3D image analysis would be useful for evaluating osteoporosis, rather than 2D image analysis, and would exclude the observer's subjection to select the target image. Although the target VOI range was arbitrary set with semi-auto-segmentation of the 3D analysis software, our results proved this target VOI set was appropriate and reproducible. More precise target volume

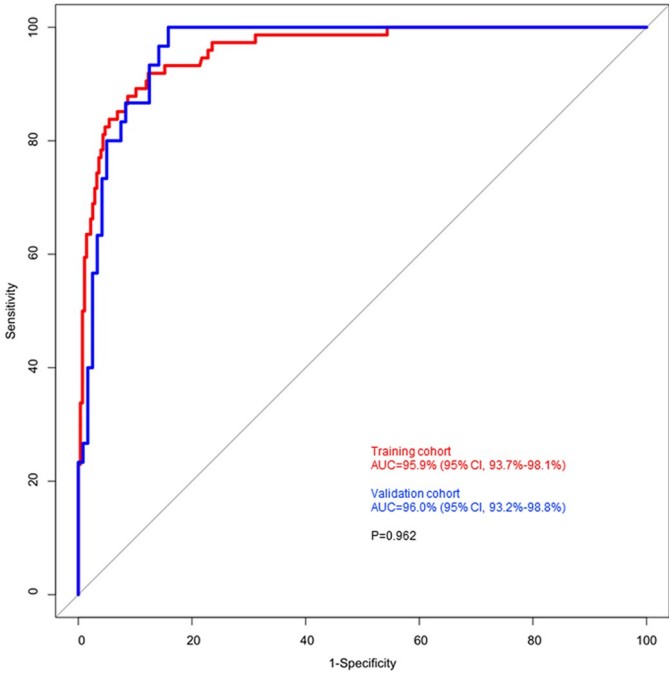

**Fig 4. Comparison of area under curves of training and validation cohorts.**

selection focusing the femoral neck or Ward's triangle would likely improve the diagnostic accuracy; however, the current VOI selection proved effective to predict osteoporosis because this method was easy, simple, and highly reproducible.

In this study, among the forty-one radiomic features, only six radiomics features were excluded because of low reproducibility. Most features showed a high reproducibility of more than 0.9 of ICC. Advantage of the 3D image software is high reproducibility [34, 35]. As the femur shows a high contrast with the surrounding tissue, automatic segmentation is relatively easy. Therefore, it is expected that automatic osteoporosis prediction can be implanted on a picture archiving and communication system or workstation of CT consoles through automatic femur segmentation and machine-learning analysis during CT acquisition and post-image processing. Furthermore, we researched the prediction of osteoporosis by applying machine learning analysis using specific radiomics features and HUHA due to the limited number of patients in this study. Recent AI research is moving to deep learning, which is the evolution of machine learning and it helps in making better precision medicine than machine learning. Deep learning is similar to machine learning, but it does not require artificial intervention. However, it requires big data to train the model otherwise it won't work as expected. If osteoporosis screening can be implemented by deep learning using wide and easily accessible plain radiographs or large number of CT images in the future, great progress can be expected in the prevention and treatment of osteoporosis.

Although this study achieved a high prediction accuracy, the major limitation was these results were obtained from a single center and single race. Osteoporosis differs according to gender and race [36]. However, our results were based on the DXA results within a 1-month interval, the only standard reference of osteoporosis diagnosis. Based on our research results, it is necessary to prove the validity through prospective multi-center and multi-ethnic studies. Osteoporosis is divided into three groups according to the DXA T-score. Our study cohort was divided into two groups: osteoporosis and non-osteoporosis groups, which consist of normal and osteopenia patients. This classification was based on the following two reasons. First, in the Korean medical insurance system, insurance coverage is applied only to patients diagnosed with osteoporosis. Second, the purpose of this study was to evaluate the predicting performance of opportunistic screening for osteoporosis using APCT. Thus, this study was focused on osteoporosis prediction. In the future, it will be necessary to evaluate whether radiomics analysis is possible to predict osteoporosis, osteopenia, and normal status accurately. Our study included a large number of patients with surveillance of breast cancer metastasis. However, in order to minimize the possibility of potential breast cancer metastasis in imaging analysis, we selected the breast cancer patients under the specific evaluation that they were those who did not find any metastases in three consecutive APCT at 6-month intervals. Give that osteoporosis is a common disease in women, the expected effect will be great if the opportunistic screening of osteoporosis is performed simultaneously in female patients diagnosed with breast cancer.

In conclusion, prediction performance of femoral osteoporosis using the machine-learning analysis with radiomics features and APCT proved high validity with more than 93% of accuracy, specificity, and negative predictive value. Overall, opportunistic screening of femoral osteoporosis with machine-learning analysis and APCT has shown high potential feasibility.

## Supporting information

**S1 Table.**
(DOCX)

**S1 Appendix.**
(DOCX)

## Acknowledgments

The dataset in this paper can be fully accessed through the following address.

"Ha, Hong Il; Lim, Hyun Kyung; Park, Sun-Young; & Han, Junhee (2021). Prediction of femoral osteoporosis using machine-learning analysis with radiomics features and abdomen-pelvic CT: A retrospective single center preliminary study. Zenodo. https://doi.org/10.5281/zenodo.4460972".

## Author Contributions

**Conceptualization:** Hong Il Ha.

**Data curation:** Hong Il Ha, Sun-Young Park.

**Formal analysis:** Hyun Kyung Lim, Hong Il Ha, Junhee Han.

**Funding acquisition:** Hyun Kyung Lim, Hong Il Ha.

**Investigation:** Hyun Kyung Lim, Hong Il Ha, Sun-Young Park.

**Methodology:** Hong Il Ha, Sun-Young Park, Junhee Han.

**Project administration:** Hong Il Ha.

**Resources:** Hong Il Ha.

**Software:** Hong Il Ha.

**Supervision:** Hong Il Ha.

**Validation:** Hong Il Ha.

**Visualization:** Hong Il Ha.

**Writing – original draft:** Hyun Kyung Lim, Hong Il Ha.

**Writing – review & editing:** Hyun Kyung Lim, Hong Il Ha, Sun-Young Park.

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
