## [Decision Letter · Decision Letter 0]

19 Jan 2021

PONE-D-20-33039

Prediction of femoral osteoporosis using machine-learning analysis with radiomics features and abdomen-pelvic CT: A retrospective single center preliminary study

PLOS ONE

Dear Dr. Ha,

Thank you for submitting your manuscript to PLOS ONE. After careful consideration, we feel that it has merit but does not fully meet PLOS ONE’s publication criteria as it currently stands. Therefore, we invite you to submit a revised version of the manuscript that addresses the points raised during the review process.

We suggest you pay special attention to address the experimental design concerns of reviewer #1.

We look forward to receiving your revised manuscript.

Kind regards,

Alfredo Vellido

Academic Editor

PLOS ONE

2. Please state whether the data utilized in this study were de-identified/anonymised before access?

'Hyun Kyung Lim received the Soonchunhyang University research fund. This fund has no specific grant number.

Hong Il Ha received fund from the DongKook Life Science. Co., Ltd., Republic of Korea (DK-IIT2019-03).

These funders had no role in study design, data collection and analysis, decision to publish, or preparation of the manuscript.'

We note that you received funding from a commercial source: DongKook Life Science Co Ltd.

Reviewers' comments:

Reviewer's Responses to Questions

**Comments to the Author**

1. Is the manuscript technically sound, and do the data support the conclusions?

Reviewer #1: Yes

Reviewer #2: Yes

2. Has the statistical analysis been performed appropriately and rigorously? 

Reviewer #1: Yes

Reviewer #2: Yes

3. Have the authors made all data underlying the findings in their manuscript fully available?

Reviewer #1: No

Reviewer #2: Yes

4. Is the manuscript presented in an intelligible fashion and written in standard English?

Reviewer #1: Yes

Reviewer #2: Yes

5. Review Comments to the Author

Reviewer #1: " Prediction of femoral osteoporosis using machine-learning analysis with radiomics features and abdomen-pelvic CT: A retrospective single center preliminary study”

It is very interesting that your academic paper evaluated the prediction performance of femoral osteoporosis using machine-learning analysis with radiomics features and APCT. This manuscript is very well written. However, there are some corrections that are essential to meet the standard for publication. Please refer to the following comments.

1. Did you evaluate the validity of the model by cross-validation? If not, please add it. If you have done so, please add it to the supplementary materials.

2. Please indicate the hyperparameters of Random Forest. Did you adjust the hyperparameters in this study? If so, please add the optimized method and result. If you haven't done so, expect to tune for this result.

3. Which data did you use when you took multiple CT scans during the survey period? Also, did any of the same patients have multiple BMD measurements? Please indicate whether the patient's data used either one or both were included in the analysis.

4. In recent years, deep learning has been attracting attention. The authors have shown very good results in machine learning. Make a comparison with deep learning and show your thoughts.

Reviewer #2: Please use the following studies in your paper and compare the results:

1) Radiomics for classification of bone mineral loss: A machine learning study

2) Magnetic resonance imaging radiomic feature analysis of radiation-induced femoral head changes in prostate cancer radiotherapy

6. PLOS authors have the option to publish the peer review history of their article (what does this mean?). If published, this will include your full peer review and any attached files.

Reviewer #1: No

Reviewer #2: **Yes: **I do not sign this review on behalf of another person

---

## [Author Response · Author response to Decision Letter 0]

29 Jan 2021

Point by point response of Reviewer comment

Response to the general comments by editor and editorial office:

1. We revised the financial disclosure section information.

We provide an amended competing interest statement as following, 

“The DongKook Life Science. Co., Ltd. has no competing interests relating to employment, consultancy, patents, products in development, marketed products, etc. These funders had no role in study design, data collection and analysis, decision to publish, or preparation of the manuscript. These do not alter our adherence to PLOS ONE policies on sharing data and materials.”

We add the amended Competing Interests Statement within our revised cover letter. 

2. We include captions of supporting information files at the end of our manuscript.

Supporting information files

S1. Table of five-fold cross validation results of random forest model.

S2. Appendix of graphical explanation of random forest model.

3. We added data availability statement in the cover letter and manuscript. 

Data Availability: The datasets in this paper can be fully accessed through the following address.

“Ha, Hong Il; Lim, Hyun Kyung; Park, Sun-Young; & Han, Junhee (2021). Prediction of femoral osteoporosis using machine-learning analysis with radiomics features and abdomen-pelvic CT: A retrospective single center preliminary study. Zenodo. https://doi.org/10.5281/zenodo.4460972”

Comment and response of Reviewer #1

1. Did you evaluate the validity of the model by cross-validation? If not, please add it. If you have done so, please add it to the supplementary materials.

We performed 5-fold cross validation and added it to supplementary material named as “S1. Table of five-fold cross validation results of random forest model”. 

2. Please indicate the hyperparameters of Random Forest. Did you adjust the hyperparameters in this study? If so, please add the optimized method and result. If you haven't done so, expect to tune for this result.

We add the tuning of hyperparameters of random forest model as follows, 

“For the best hyperparameter tuning, five-fold cross validation using random search was performed and the result was summarized in supplementary file (S1. Table of five-fold cross validation results of random forest model). The hyperparameter of our RF algorithm were as follows: mtry=3, minimum nodal size=11, and splitrule=extratrees. The best model was selected and validated in the test cohort. Further explanation of RF model and features was summarized in supplementary file (S2. Appendix of graphical explanation of random forest model).

3. Which data did you use when you took multiple CT scans during the survey period? Also, did any of the same patients have multiple BMD measurements? Please indicate whether the patient's data used either one or both were included in the analysis.

No. 500 APCT cases were obtained from 500 non-duplicated patients and all patients had matched BMD test results. To avoid confusion, we add following sentence. 

“There were no duplicate patients enrolled.”

4. In recent years, deep learning has been attracting attention. The authors have shown very good results in machine learning. Make a comparison with deep learning and show your thoughts.

We add following paragraph in discussion section. 

“Furthermore, we researched the prediction of osteoporosis by applying machine learning analysis using specific radiomics features and HUHA due to the limited number of patients in this study. Recent AI research is moving to deep learning, which is the evolution of machine learning and it helps in making better precision medicine than machine learning. Deep learning is similar to machine learning, but it does not require artificial intervention. However, it requires big data to train the model otherwise it won’t work as expected. If osteoporosis screening can be implemented by deep learning using wide and easily accessible plain radiographs or large number of CT images in the future, great progress can be expected in the prevention and treatment of osteoporosis.” 

Comment and response of Reviewer #2 

Please use the following studies in your paper and compare the results:

1) Radiomics for classification of bone mineral loss: A machine learning study

Rastegar S, Vaziri M, Qasempour Y, Akhash MR, Abdalvand N, Shiri I, Abdollahi H, Zaidi H. Radiomics for classification of bone mineral loss: A machine learning study. Diagn Interv Imaging. 2020 Sep;101(9):599-610. doi: 10.1016/j.diii.2020.01.008. Epub 2020 Feb 4. PMID: 32033913.

2) Magnetic resonance imaging radiomic feature analysis of radiation-induced femoral head changes in prostate cancer radiotherapy

Abdollahi H, Mahdavi SR, Shiri I, Mofid B, Bakhshandeh M, Rahmani K. Magnetic resonance imaging radiomic feature analysis of radiation-induced femoral head changes in prostate cancer radiotherapy. J Cancer Res Ther. 2019 Mar;15(Supplement):S11-S19. doi: 10.4103/jcrt.JCRT_172_18. PMID: 30900614.

We thoroughly reviewed the papers presented. However, we found that the experimental methods and the subject of the study differed greatly from our study. Therefore, the comparison and discussion between the results of these papers and the results of our research does not seem to fit the subject of this research theme. However, since both of these papers are valuable papers that analyzed femoral structural changes through radiomics analysis, we cite these papers as references to the radiomics analysis description mentioned in the introduction as #14 and #15, respectively.

---

## [Decision Letter · Decision Letter 1]

2 Feb 2021

PONE-D-20-33039R1

Prediction of femoral osteoporosis using machine-learning analysis with radiomics features and abdomen-pelvic CT: A retrospective single center preliminary study

PLOS ONE

Dear Dr. Ha,

Thank you for submitting your manuscript to PLOS ONE. After careful consideration, we feel that it has merit but does not fully meet PLOS ONE’s publication criteria as it currently stands. Therefore, we invite you to submit a revised version of the manuscript that addresses the points raised during the review process.

We look forward to receiving your revised manuscript.

Kind regards,

Alfredo Vellido

Academic Editor

PLOS ONE

Additional Editor Comments (if provided):

Dear authors, one of the reviewers is already happy with the current version of your manuscript, but you still need to carefully address the concerns raised by the other reviewer.

Reviewers' comments:

Reviewer's Responses to Questions

**Comments to the Author**

1. If the authors have adequately addressed your comments raised in a previous round of review and you feel that this manuscript is now acceptable for publication, you may indicate that here to bypass the “Comments to the Author” section, enter your conflict of interest statement in the “Confidential to Editor” section, and submit your "Accept" recommendation.

Reviewer #1: All comments have been addressed

Reviewer #2: All comments have been addressed

2. Is the manuscript technically sound, and do the data support the conclusions?

Reviewer #1: (No Response)

Reviewer #2: Yes

3. Has the statistical analysis been performed appropriately and rigorously? 

Reviewer #1: No

Reviewer #2: Yes

4. Have the authors made all data underlying the findings in their manuscript fully available?

Reviewer #1: Yes

Reviewer #2: Yes

5. Is the manuscript presented in an intelligible fashion and written in standard English?

Reviewer #1: Yes

Reviewer #2: Yes

6. Review Comments to the Author

Reviewer #1: Thank you for giving me this opportunity to re-review your revised manuscript.

I am happy that almost all of the suggested corrections have been made.

Please refer only to the minor comments below.

Thank you for spending so much time for revised manuscript.

1. Did you evaluate the validity of the model by cross-validation? If not, please add it. If you have done so, please add it to the supplementary materials.

The authors explained that they performed cross-validation to fine-tune the hyperparameters.

Cross-validation is generally an analysis performed to ensure generalization.

Is the result of this analysis of the gini coefficient etc. the holdout method?

Please show the results of cross-validation.

In addition, please show me how to split data for cross-validation. Results may vary depending on the proportion of test data.

Additionally, did you change the hyperparameter adjustment from the default value? Also show the changes from the initial values of hyperparameters.

Reviewer #2: It is an interesting study on the bone radiomics analysis.

7. PLOS authors have the option to publish the peer review history of their article (what does this mean?). If published, this will include your full peer review and any attached files.

Reviewer #1: No

Reviewer #2: No

---

## [Author Response · Author response to Decision Letter 1]

2 Feb 2021

Response to Reviewers

Reviewer #1: Thank you for giving me this opportunity to re-review your revised manuscript.

I am happy that almost all of the suggested corrections have been made.

Please refer only to the minor comments below.

Thank you for spending so much time for revised manuscript.

1. Did you evaluate the validity of the model by cross-validation? If not, please add it. If you have done so, please add it to the supplementary materials.

The authors explained that they performed cross-validation to fine-tune the hyperparameters.

Cross-validation is generally an analysis performed to ensure generalization.

Is the result of this analysis of the gini coefficient etc. the holdout method?

Please show the results of cross-validation.

In addition, please show me how to split data for cross-validation. Results may vary depending on the proportion of test data.

Additionally, did you change the hyperparameter adjustment from the default value? Also show the changes from the initial values of hyperparameters

We attach the original results of 5-folds cross validation confusion matrix. Supplementary file named “S1. Table” is summary of this results. 

 We divided the total 500 cases into 70% of the train set and 30% of the validation set. And hyperparameter tuning was performed while performing 5-fold cross validation on the train set. The test set result was the validation result of RF model using a tuned hyperparameter by 5-fold cross validation. 

In addition, we attach the our R-statistical code. This code contains all information of our random forest model about data split, train and test set composition and hyperparameter searching algorithm, etc. This is critical information of this research because we have 2nd phase study related this research. So, we can't publish this "R-code data". However, we already submit our full data set so anyone can test and validate our research. 

A. Five-fold cross validation confusion matrix results 

1) Fold 1.

2) Fold 2

3) Fold 3

4) fold 4

5) Fold 5

B. R- code for random forest model 

library(randomForest)

library(MASS)

library(caret)

library(dplyr)

library(caTools)

library(e1071)

library(tidyverse)

library(tictoc)

library(janitor)

library(doSNOW)

library(ranger)

library(BradleyTerry2)

library(randomForestExplainer)

setwd()

getwd()

data <- read.csv(file="data.csv", header=T)

data["Dx"] <- as.factor(data$Dx)

levels(data$Dx)=c("negative","positive")

set.seed(123)

trainIndex <- createDataPartition(data$Dx, p=.7, list=FALSE, times=1)

train = data[trainIndex,]

test = data[-trainIndex,]

print(table(train$Dx) / nrow(train))

print(table(test$Dx) / nrow(test))

# K-folds 

set.seed(123)

cv_folds_lst <- createFolds(train$Dx, k=5, list=FALSE)

set.seed(123)

cv_folds <- createFolds(train$Dx, k=5, list=TRUE)

ranger_tune_grid <- expand.grid(

 .mtry = c(2:32),

 .splitrule = c("gini","extratrees"),

 .min.node.size = c(2:30)

)

fit_ctrl <-trainControl(method = "repeatedcv",

 number = 5,

 repeats = 5,

 index = cv_folds,

 summaryFunction=twoClassSummary,

 classProbs = TRUE,

 verboseIter = TRUE)

set.seed(123)

gc_grid_ranger_model <- train(Dx ~., train,

 method = "ranger",

 metric = "AUC",

 preProcess = c("zv", "center", "scale", "spatialSign"),

 tuneGrid = ranger_tune_grid,

 #tuneLength = 15,

 trControl = fit_ctrl)

gc_grid_ranger_model

# The final values used for the model were mtry = 10, splitrule = extratrees and min.node.size = 3.

fit_ctrl <- trainControl(method = "adaptive_cv", 

 number = 5,

 repeats = 1,

 index = cv_folds, 

 search = "random",

 adaptive = list(min = 3, alpha = 0.05, method = "BT", complete = FALSE),

 summaryFunction = twoClassSummary,

 classProbs = TRUE, 

 verboseIter = TRUE)

gc_ranger_model <- train(Dx ~., train,

 method = "ranger",

 metric = "Sens",

 preProcess = c("zv", "center", "scale", "spatialSign"),

 trControl = fit_ctrl,

 tuneLength = 7)

gc_ranger_model

for (ntrees in c(25, 50, 100, 250, 500, 750, 1000, 2000)){

 print(ntrees)

 acc_vec <- c()

 for (idx in c(1, 2, 3, 4, 5)){

 num <- numeric(idx)

 #print(idx)

 cv_train <- train[cv_folds_lst != idx,]

 cv_validation <- train[cv_folds_lst == idx,]

 X_validation <- cv_validation[, -33]

 y_validation <- cv_validation[, 33]

formula.init <- "Dx ~ ."

formula.init <- as.formula(formula.init)

 set.seed(123)

 rf.model <- randomForest(formula=formula.init, data=cv_train, proximity=T, ntree=ntrees,mtry = 3

 , splitrule="extratrees", min.node.size=11)

 rf.predictions <- predict(rf.model,X_validation, type="class")

 #print(rf.model)

 #print(confusionMatrix(data=rf.predictions, reference =y_validation, positive="No"))

 acc_vec <- append(acc_vec,mean(rf.predictions == y_validation))

 }

 print(acc_vec)

 print(mean(acc_vec))

}

for (idx in c(1, 2, 3, 4, 5)){

 num <- numeric(idx)

 print(idx)

 cv_train <- train[cv_folds_lst != idx,]

 cv_validation <- train[cv_folds_lst == idx,]

 X_validation <- cv_validation[, -33]

 y_validation <- cv_validation[, 33]

formula.init <- "Dx ~ ."

formula.init <- as.formula(formula.init)

 rf.model <- randomForest(formula=formula.init, data=cv_train, proximity=T,ntree=500,mtry = 3, 

 splitrule="extratrees", min.node.size=11)

 rf.predictions <- predict(rf.model,X_validation, type="class")

 print(confusionMatrix(data=rf.predictions, reference =y_validation, positive="positive"))

}

X_test <- test[, -33]

y_test <- test[, 33]

formula.init <- "Dx ~ ."

formula.init <- as.formula(formula.init)

set.seed(123)

rf.model <- randomForest(formula=formula.init, data=train, proximity=T,ntree=500,mtry =3, splitrule="extratrees", min.node.size=11)

rf.predictions <- predict(rf.model,X_test, type="class")

confusion <- confusionMatrix(data=rf.predictions, reference =y_test, positive="?缺")

rf.model$importance

varImpPlot(rf.model)

X_test <- test[, -33]

y_test <- test[, 33]

test_predict <- predict(rf.model,X_test, type="class")

levels(y_test)=c(0,1)

levels(test_predict)=c(0,1)

y_test <- as.numeric(levels(y_test))[y_test]

test_predict <- as.numeric(levels(test_predict))[test_predict]

train_model <- rf.model$predicted

train_model <- as.factor(as.numeric(train_model))

levels(train_model)=c(0,1)

library(pROC)

par(pty="s")

trainROC <- roc(y_train ~ as.numeric(levels(train_model))[train_model],plot=TRUE,print.auc=TRUE,col="blue",

 lwd =4,legacy.axes=TRUE,main="ROC Curves", percent=TRUE)

 #ylab="False Positive Percentage", xlab = "True Positive Percentage"

## Setting levels: control = 0, case = 1

## Setting direction: controls < cases

testROC <- roc(y_test ~ test_predict,plot=TRUE,print.auc=TRUE,col="red",lwd = 4,print.auc.y=45,legacy.axes=TRUE,add = TRUE, percent=TRUE)

## Setting levels: control = 0, case = 1

## Setting direction: controls < cases

legend("bottomright",legend=c("Training set","Test set"),col=c("blue","red"),lwd=4)

library(ROCR)

library(pROC)

library(randomForest)

#data dependent variable set

set.seed(123)

train$Dx = as.factor(train$Dx)

data1.rf <- randomForest(Dx ~., data=train, proximity=T,ntree=500,mtry = 3, splitrule="extratrees", min.node.size=11)

test$Dx = as.factor(test$Dx)

data2.rf <- randomForest(Dx ~., data=test,proximity=T,ntree=500,mtry = 3, splitrule="extratrees", min.node.size=11)

par(pty="s")

set.seed(123)

require(pROC)

rf.roc1 <-roc(train$Dx,data1.rf$votes[,2], plot=TRUE, legacy.axes=TRUE, percent = TRUE, 

 xlab="1-Specificity",ylab='Sensitivity',col='#FF0000',

 lwd=4,

 print.auc=TRUE,print.auc.y=25)

rf.roc2 <-roc(test$Dx,data2.rf$votes[,2], plot=TRUE, legacy.axes=TRUE, percent = TRUE, 

 xlab="1-Specificity",ylab='Sensitivity',col='#0000FF',

 lwd=4,

 print.auc=TRUE,add=TRUE, print.auc.y=20)

ci(rf.roc1)

ci(rf.roc2)

roc.test(rf.roc1, rf.roc2, method=c("delong", "bootstrap",

 "venkatraman", "sensitivity", "specificity"), sensitivity = NULL,

 specificity = NULL, alternative = c("two.sided", "less", "greater"),

 paired=NULL, reuse.auc=TRUE, boot.n=2000, boot.stratified=TRUE,

 ties.method="first", progress=getOption("pROCProgress")$name,

 parallel=FALSE)

library(randomForestExplainer)

library(randomForest)

library(tidyverse)

set.seed(123)

forest <- randomForest::randomForest(Dx ~ ., data = data, localImp = TRUE, proximity=T,mtry = 3, splitrule="extratrees", min.node.size=11)

suppressPackageStartupMessages(suppressMessages(suppressWarnings(explain_forest(forest, interactions = TRUE))))

---

## [Decision Letter · Decision Letter 2]

5 Feb 2021

Prediction of femoral osteoporosis using machine-learning analysis with radiomics features and abdomen-pelvic CT: A retrospective single center preliminary study

PONE-D-20-33039R2

Dear Dr. Ha,

We’re pleased to inform you that your manuscript has been judged scientifically suitable for publication and will be formally accepted for publication once it meets all outstanding technical requirements.

Kind regards,

Alfredo Vellido

Academic Editor

PLOS ONE

Additional Editor Comments (optional):

Reviewers' comments:

Reviewer's Responses to Questions

**Comments to the Author**

1. If the authors have adequately addressed your comments raised in a previous round of review and you feel that this manuscript is now acceptable for publication, you may indicate that here to bypass the “Comments to the Author” section, enter your conflict of interest statement in the “Confidential to Editor” section, and submit your "Accept" recommendation.

Reviewer #1: All comments have been addressed

2. Is the manuscript technically sound, and do the data support the conclusions?

Reviewer #1: Yes

3. Has the statistical analysis been performed appropriately and rigorously? 

Reviewer #1: Yes

4. Have the authors made all data underlying the findings in their manuscript fully available?

Reviewer #1: Yes

5. Is the manuscript presented in an intelligible fashion and written in standard English?

Reviewer #1: Yes

6. Review Comments to the Author

Reviewer #1: Thank you for giving me this opportunity to re-review your revised manuscript.

I am happy that all of the suggested corrections have been made.

Thank you for spending so much effort.

7. PLOS authors have the option to publish the peer review history of their article (what does this mean?). If published, this will include your full peer review and any attached files.

Reviewer #1: No

---

## [Editor Report · Acceptance letter]

10 Feb 2021

PONE-D-20-33039R2 

Prediction of femoral osteoporosis using machine-learning analysis with radiomics features and abdomen-pelvic CT: A retrospective single center preliminary study 

Dear Dr. Ha:

I'm pleased to inform you that your manuscript has been deemed suitable for publication in PLOS ONE. Congratulations! Your manuscript is now with our production department. 

Kind regards, 

on behalf of

Dr. Alfredo Vellido 

Academic Editor

PLOS ONE